# Effects of turmeric (*Curcuma longa*) supplementation on glucose metabolism in diabetes mellitus and metabolic syndrome: An umbrella review and updated meta-analysis

**Thanika Pathomwichaiwat**[1], **Peerawat Jinatongthai**[2]\*, **Napattaoon Prommasut**[3], **Kanyarat Ampornwong**[3], **Wipharak Rattanavipanon**[3], **Surakit Nathisuwan**[3]\*, **Ammarin Thakkinstian**[4]

**1** Faculty of Pharmacy, Department of Pharmaceutical Botany, Mahidol University, Bangkok, Thailand,
**2** Faculty of Pharmaceutical Sciences, Pharmacy Practice Division, Ubon Ratchathani University, Ubon Ratchathani, Thailand, **3** Faculty of Pharmacy, Department of Pharmacy, Mahidol University, Bangkok, Thailand, **4** Faculty of Medicine Ramathibodi Hospital, Department of Clinical Epidemiology and Biostatistics, Mahidol University, Bangkok, Thailand

\* peerawat.j@ubu.ac.th (PJ); surakit.nat@mahidol.ac.th (SN)

## Abstract

### Aims

This study aims to comprehensively review the existing evidence and conduct analysis of updated randomized controlled trials (RCTs) of turmeric (*Curcuma longa*, CL) and its related bioactive compounds on glycemic and metabolic parameters in patients with type 2 diabetes (T2DM), prediabetes, and metabolic syndrome (MetS) together with a sub-group analysis of different CL preparation forms.

### Methods

An umbrella review (UR) and updated systematic reviews and meta-analyses (SRMAs) were conducted to evaluate the effects of CL compared with a placebo/standard treatment in adult T2DM, prediabetes, and MetS. The MEDLINE, Embase, The Cochrane Central Register of Control Trials, and Scopus databases were searched from inception to September 2022. The primary efficacy outcomes were hemoglobin A1C (HbA1C) and fasting blood glucose (FBG). The corrected covered area (CCA) was used to assess overlap. Mean differences were pooled across individual RCTs using a random-effects model. Subgroup and sensitivity analyses were performed for various CL preparation forms.

### Results

Fourteen SRMAs of 61 individual RCTs were included in the UR. The updated SRMA included 28 studies. The CCA was 11.54%, indicating high overlap across SRMAs. The updated SRMA revealed significant reduction in FBG and HbA1C with CL supplementation, obtaining a mean difference (95% confidence interval [CI]) of −8.129 (−12.175, −4.084) mg/dL and −0.134 (−0.304, −0.037) %, respectively. FBG and HbA1C levels decreased with all

**Data Availability Statement:** All relevant data are within the manuscript and its supporting information files.

**Funding:** This study was supported by the National Research Council of Thailand (N42A640323). The funders had no role in study design, data collection and analysis, decision to publish, or preparation of the manuscript.

**Competing interests:** The authors have declared that no competing interests exist.

**Abbreviations:** CCA, corrected covered area; CL, *Curcuma longa*; CRP, C-reactive protein; FBG, fasting blood glucose; hs-CRP, high sensitivity C-reactive protein; MA, meta-analysis; SMD, standardized mean difference; SRMAs, systematic reviews and meta-analyses; TC, total cholesterol; TG, triglycerides; UMD, unstandardized mean difference; UR, umbrella review.

CL preparation forms as did other metabolic parameters levels. The results of the sensitivity and subgroup analyses were consistent with those of the main analysis.

## Conclusion

CL supplementation can significantly reduce FBG and HbA1C levels and other metabolic parameters in T2DM and mitigate related conditions, including prediabetes and MetS.

## Trial registration

PROSPERO (CRD42016042131).

## Introduction

Type 2 diabetes mellitus (T2DM) has rapidly become a leading global health burden among non-communicable diseases (NCD), given its serious complications, including heart disease, cerebrovascular disease, renal failure, limb amputation and blindness [1]. Over 437.9 million people were living with T2DM in 2019, 80% of whom were in low- and middle-income countries (LMICs). The disease accounts for approximately 1.5 million deaths and 66.3 million disability-adjusted life years worldwide [2]. The rise in T2DM cases along with the public health burden has been much more prominent in the developing than in the developed countries, which may be due to limited healthcare resources, less organized and immature health services, lower health literacy, and limited access to advanced treatment options [3,4].

While the management of T2DM requires multiple interventions ranging from lifestyle modification, social policy, health system management, to public education, access to treatment is always the cornerstone of disease control [5–8]. The accessibility of advanced treatments among LMICs is limited due to the high cost of newer treatments [9]; thus, many LMICs, especially those in tropical regions, have attempted to uncover the pharmacological properties of native plants that have long been used for medicinal purposes [10]. Some plants have been extensively studied for diabetes, *Curcuma longa* L. (CL), a member of the Zingiberaceae family, has been used as both food and medicine [11–13]. Curcumin and its related compounds, such as curcuminoids and essential oils, have been extensively studied for their antidiabetic effects ranging from molecular mechanisms, cellular signaling, *in vitro/in vivo* studies and randomized, controlled trials (RCTs) [12,14,15]. Several systematic reviews and meta-analyses (SRMAs) of RCTs [15–20] have been performed to assess the anti-diabetic effects of curcumin, but the results obtained were conflicting due to different populations, CL preparation forms, duration of use and outcomes.

An umbrella review (UR) can summarize the findings of these SRMAs and evaluate the quality of evidence, as well as assess the robustness of key findings [21]. Therefore, the present UR of SRMAs of RCTs was conducted to comprehensively review the existing evidence on the effects of turmeric preparations and its related bioactive compounds on glycemic and metabolic parameters in T2DM, prediabetes, and MetS. In addition, an updated MA was also performed by including new RCTs to perform a comprehensive analysis of glycemic and metabolic markers, along with a subgroup analysis of different CL preparation forms.

## Methods

### Study design

The protocols for the UR and updated MA have been registered with the PROSPERO registry (CRD42016042131). This study has complied with the recommendations of the Preferred

Reporting Items for Systematic Reviews and Meta-Analyses (PRISMA) guidelines for reporting SRMAs [22].

## Search strategy

For the UR, a literature search for SRMAs was conducted without language restrictions on the MEDLINE (PubMed), EMBASE, Cochrane Central Register of Control Trials, and Scopus databases from inception to September 2022. For the updated MA, we retrieved RCTs from the most recent SRMAs conducted between March and September 2022. The search algorithms and strategies are provided in Figs 1 and 2 and S1 Table in S1 File.

## Study selection

Two reviewers (T.P. and P.J.) independently screened the titles and abstracts of the retrieved articles to identify potentially relevant studies. The full texts were retrieved if no decision the selection could be made. Any discrepancy was resolved by a third reviewer (S.N.). Studies were

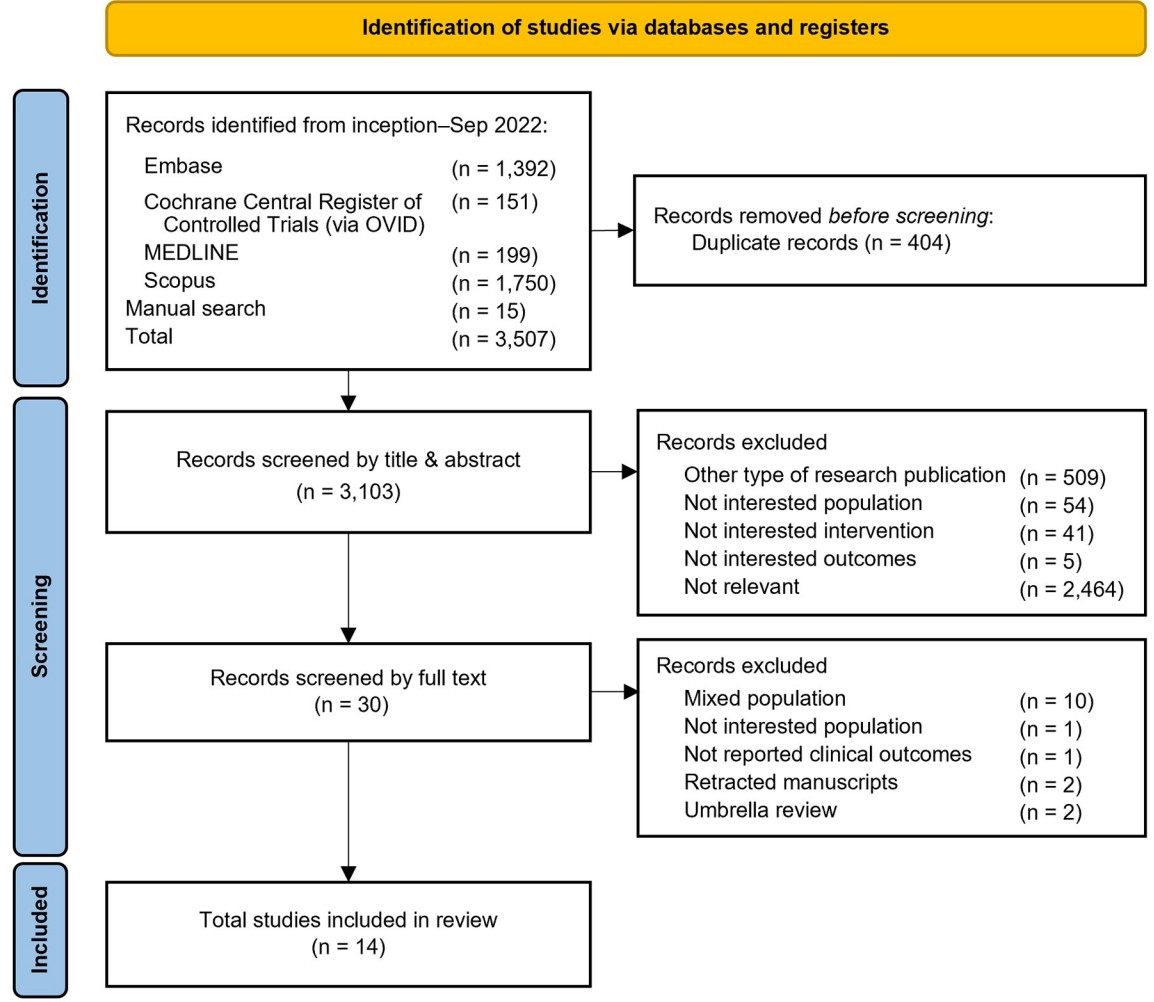

**Fig 1. Flow diagram of included studies in umbrella review.**

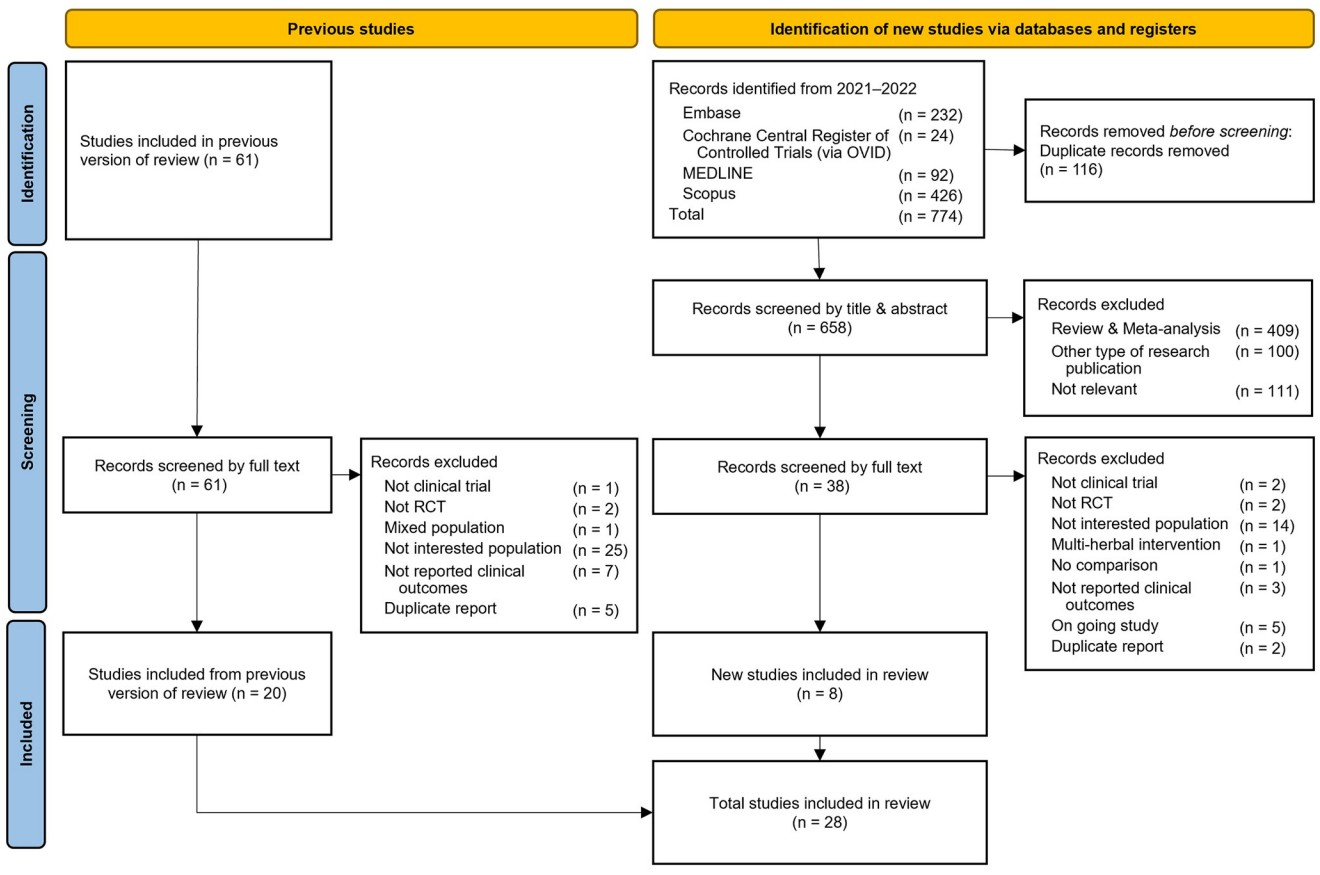

**Fig 2. Flow diagram of included studies in updated meta-analysis.**

eligible for review if they met the following criteria: (i) SRMAs of RCTs or individual RCTs; (ii) studied in adult patients with T2DM, prediabetes, or MetS; (iii) investigated the effects of CL supplementation in addition to the standard treatment and compared with the placebo or standard treatment; (iv) compared blood glucose parameters (i.e., fasting blood glucose [FPG], glycated hemoglobin A1C [HbA1C]).

## Types of interventions

CL supplementation could be administered in any of the following forms: 1) a whole preparation rich in phytochemicals, e.g., dry rhizome or standardized CL powder; 2) an extract preparation containing only portion of the phytochemicals found in CL, e.g., a standardized curcuminoid extract; and 3) a bioavailability-enhanced preparation, a modified CL preparation containing curcumin or curcuminoids (e.g., nanomicelle curcumin, liposome preparations [e.g., phospholipid complex, phytosome, phosphatidylcholine, phosphatidylserine, or phospholipid curcumin]) or any CL preparation with a low dose of piperine (S2 Table in S1 File).

## Outcomes of interest

The primary outcomes were the HbA1C and FBG levels measured 4–16 weeks after receiving CL supplementation. Secondary outcomes were lipid profiles: low-density lipoprotein

cholesterol (LDL-C), high-density lipoprotein cholesterol (HDL-C), total cholesterol (TC), and triglycerides (TG). The other selected outcomes were also collected including systolic blood pressure (SBP), diastolic blood pressure (DBP), homeostatic model assessment of insulin resistance (HOMA-IR), uric acid and high sensitivity C-reactive protein (hs-CRP).

### Data extraction and quality assessment

Relevant data were extracted by two reviewers (N.P. and K.A.) using a standardized extraction form. The extracted data included study/patient characteristics, interventions, outcomes, and other relevant findings. For outcomes, pooled effect sizes of each outcome (i.e., unstandardized mean difference [UMD]; standardized mean difference [SMD]) along with p-values and corresponding 95% confidence intervals (CI) for both postintervention values and the change from baseline were extracted.

The methodological quality of the studies was independently assessed by at least two reviewers (T.P., P.J., or W.R.) using the AMSTAR 2 tool [23] for SRMAs and the Cochrane risk-of-bias tool (RoB 2) [24] for individual RCTs. All extracted data were cross-checked by at least two other reviewers (P.J., W.R., and A.T.) and discrepancies resolved by consensus.

### Quality of evidence

The quality of evidence was evaluated by W.R. and P.J. using GRADEpro® GDT software online version [25] based on five domains, namely, the risk of bias, inconsistency, indirectness, imprecision, and publication bias. The quality of evidence was graded as high, moderate, low, or very low.

### Data synthesis and statistical analysis

For the UR, the overlap of primary studies was assessed across the included SRMAs [26,27] and assessed using Graphical Representation of Overlap for OVErviews (GROOVE) [28].

For the updated MA, the effect sizes (i.e., UMD, SMD) along with 95% CIs of the glucose and metabolism outcomes were estimated and pooled across studies using the DerSimonian and Laird random-effects model [29]. Heterogeneity and publication bias were assessed using the $I^2$ statistic and Egger's test [30], respectively.

Pre-specified subgroup analyses were performed for the types of patients (diabetes, prediabetes, and MetS), CL preparation forms (whole powder, extract, and bioavailability-enhanced preparation), and baseline characteristics of patients (including mean body mass index [BMI], lipid and lipoprotein profiles, blood pressure, and blood sugar). Additional post-hoc analyses were also performed for the CL preparation forms. For the sensitivity analyses, small-sized trials (<25th percentile) [31] and trials with a high risk of bias were excluded. All analyses were performed by P.J., T.P., and W.R. using STATA® version 17.0 (StataCorp, College Station, Texas, USA) along with the self-programmed STATA® for meta-analysis described elsewhere [32]. A.T. and S.N. provided technical analysis advice. A p-value ≤ 0.05 was considered statistically significant.

## Results

### Umbrella review

**Identification and selection of SRMAs and individual RCTs.** Of 3,507 identified studies, 14 SRMAs [15–20,33–40] containing 61 individual RCTs were eligible in the UR (Fig 1). For the updated MA, individual RCTs were screened from two sources. First, of the 61 RCTs in 14 SRMAs, 41 studies were excluded since most of them were conducted in non-interested

patients leaving 20 studies that met the inclusion criteria. Second, of 774 RCTs retrieved from the updated search, only 8 studies met the inclusion criteria. In total 28 RCTs were included in the updated MA (Fig 2).

**Description of SRMAs.**   All 14 SRMAs [15–20,33–40] were pairwise MAs, published in 2015–2022 and included 5–26 RCTs with sample sizes of 290–1,790 patients (S3 Table in S1 File). The outcome measures were pooled using either SMD (6 studies) or UMD (9 studies). As for the CL preparation forms, whole, extract and bioavailability-enhanced preparations were evaluated in 9, 10 and 13 studies, respectively. Most SRMAs reported the outcomes at 8–12 weeks. Additional details on each SRMA are provided in S3 Table in S1 File.

**Quality of SRMAs.**   All 14 included SRMAs [15–20,33–40] were graded with critically low quality based on the AMSTAR-2 rating (S4 Table in S1 File) due to missing information on the justification of excluded studies (100%), the funding sources of included studies (100%), missing consideration for the risk of bias when interpreting the results of the review (90.91%), missing assessment of the impact of risk of bias on study results (72.73%), and a lack of established review methods (63.64%).

**Degree of overlap in SRMAs.**   The degree of overlap of individual included RCTs across SRMAs based on the study-citation matrix is shown in S17 Fig in S1 File. The corrected covered area (CCA) score was 11.54% indicating a high degree of overlap. Thirty-one of 91 RCTs were included in multiple SRMAs, indicating limited incremental information in these SRMAs.

**Primary efficacy outcomes.**   Nine SRMAs [15,17–19,33,36,37,39,40] reported changes in FBG after CL administration through various preparations. Of these, 8 SRMAs [15,17–19,33,36,37,39,40] found significant reductions with SMDs of –0.38 to –1.09 mg/dL and UMDs of –8.85 to –27.07 mg/dL. Nine SRMAs [15,16,18,19,35–37,39,40] reported changes in HbA1C, 8 [15,16,18,19,36,37,39,40] found significant reductions with SMDs of –0.42 to –1.06 and UMDs of –0.41% to –0.85% (Table 1). In addition, 3 of 5 SRMAs found significant reductions in HOMA-IR after CL administration with SMDs of –0.28 to –0.8 (Table 1).

**Secondary efficacy outcomes.**   Among 7 SRMAs [15,16,33,36–38,40] reporting changes in LDL-C and TC, 4 [16,37,38,40] reported a significant reduction in LDL-C with an SMD of –0.28 and UMDs of –6.33 to –11.19 mg/dL, whereas 4 studies [15,16,38,40] found a significant reduction in TC with an SMD of –0.3 and UMDs of –8.91 to –12 mg/dL. Among 8 SRMAs [15–17,33,36–38,40] reporting changes in TG and HDL-C, 6 [16,17,36–38,40] found a significant reduction in TG with SMDs of –0.57 to –0.85 and UMDs of –13.99 to –33.66 mg/dL. No studies found a significant increase in HDL-C (Table 1). Several SRMAs [16,17,20,33–35,40] also reported changes in BMI, blood pressure, CRP and hs-CRP after CL administration. Results consistently showed a reduction in outcomes with CL administration but with limited significance (Table 1).

## Updated meta-analysis

**Baseline characteristics.**   Twenty-eight RCTs [20,41–72] involving 2,362 patients were included in the updated MA. The details and baseline characteristics are shown in S6 and S7 Tables in S1 File, respectively. All studies compared one to two CL forms of supplementation with placebo. Sixteen [41,42,45,49,54,55,58–62,64–66,68,70,71], six [20,43,46,47,52,63,67,72], and three [48,56,57,69] studies involved patients with T2DM, MetS, and prediabetes, respectively, whereas three studies [44,50,51,53] involved a mix of these three patient types. The mean age range was 34.5–70.0 years and 34.2–69 years in the CL and placebo groups, respectively, whereas the mean BMI range 23.4–31.7 kg/m$^2$ and 22.8–31.9 kg/m$^2$, respectively.

**CL preparation forms and outcomes.**   Regarding CL preparation forms, 11 [42,43,48,49,54,56,57,61,62,70–72], 11 [20,45–47,52,53,60,63–69], and 7

**Table 1. Difference of post-intervention values and change from baseline for glycemic and metabolic outcomes between *Curcuma longa* (CL) supplementation and control group from previous SRMAs.**

| SRMAs | RCTs (N) | Patients (N) | Type of patients | Outcome measurement | Mean difference (95% CI) | $I^2$ | Type of mean difference | Analysis model |
|---|---|---|---|---|---|---|---|---|
| **FBG** | | | | | | | | |
| Ashtary-Larky, D, 2021 [33] | 3 | 76 | MetS | Change from baseline | -28.29 (-63.34, 6.76) | 69 | UMD | Random effect |
| | 3 | 150 | T2DM | Change from baseline | **-27.07 (-39.61, -14.52)** | 54 | UMD | Random effect |
| Azhdari M, 2019 [17] | 5 | 359 | MetS | Change from baseline | **-9.18 (-16.7, -1.66)** | 90 | UMD | Random effect |
| de Melo, 2018 [18] | 8 | 923 | pre-DM, T2DM, MetS | Post-intervention value | **-13.86 (-21.17, -6.56)** | 74 | UMD | Random effect |
| Huang, J, 2019 [19]- | 5 | 509 | T2DM | Change from baseline | **-0.681 (-1.067, -0.294)** | 75 | SMD | Random effect |
| | 8 | 556 | NAFLD/MetS | Change from baseline | **-0.149 (-0.348, 0.049)** | 26 | SMD | Random effect |
| Tabrizi, R, 2018 [36] | 7 | 662 | T2DM | Change from baseline | **-1.09 (-1.91, -0.27)** | 95 | SMD | Random effect |
| Tian, J, 2022 [37] | 9 | 565 | T2DM | Change from baseline | **-8.85 (-14.4, -3.29)** | 41 | UMD | Random effect |
| Yuan, F, 2022 [39] | 11 | 1,445 | T2DM | Change from baseline | **-12.88 (-17.49, -8.28)** | 80 | UMD | Random effect |
| Zhang, T, 2021 [15] | 5 | 564 | T2DM | Change from baseline | -0.28 (-0.62, 0.06) | 72 | SMD | Random effect |
| Zheng, ZH, 2021 [40] | 4 | 316 | T2DM | Post-intervention value | **-14.49 (-21.2, -7.79)** | 34 | UMD | Fixed effect |
| **HbA1c** | | | | | | | | |
| Altobelli, E, 2021 [16] | 5 | 333 | T2DM | Change from baseline | **-0.42 (-0.72, -0.11)** | 42 | SMD | Random effect |
| de Melo, 2018 [18] | 7 | 797 | pre-DM, T2DM, MetS | Post-intervention value | **-0.54 (-1.09, -0.002)** | 90 | UMD | Random effect |
| Huang, J, 2019 [19] | 3 | 223 | NAFLD/MetS | Change from baseline | -0.244 (-0.854, 0.366) | 81 | SMD | Random effect |
| | 5 | 478 | T2DM | Change from baseline | **-0.455 (-0.713, -0.198)** | 45 | SMD | Random effect |
| Macena, ML, 2022 [35] | 1 | 46 | T2DM | Change from baseline | 0.96 (-0.28, 2.20) | NR | SMD | Random effect |
| Tabrizi, R, 2018 [36] | 6 | 583 | T2DM | Change from baseline | **-1.06 (-1.51, -0.60)** | 83 | SMD | Random effect |
| Tian, J, 2022 [37] | 9 | 565 | T2DM | Change from baseline | **-0.54 (-0.81, -0.27)** | 65 | UMD | Random effect |
| Yuan, F, 2022 [39] | 10 | 1,405 | T2DM | Change from baseline | **-0.41 (-0.56, -0.26)** | 73 | UMD | Random effect |
| Zhang, T, 2021 [15] | 5 | 524 | T2DM | Change from baseline | **-0.7 (-0.87, -0.54)** | 0 | UMD | Fixed effect |
| Zheng, ZH, 2021 [40] | 3 | 272 | T2DM | Post-intervention value | **-0.85 (-1.16, -0.54)** | 0 | UMD | Fixed effect |
| **HOMA-IR** | | | | | | | | |
| Altobelli, E, 2021 [16] | 4 | 432 | T2DM | Change from baseline | **-0.41 (-0.66, -0.22)** | 0 | SMD | Random effect |
| de Melo, 2018 [18] | 3 | 531 | pre-DM, T2DM, DLP | Post-intervention value | -1.26 (-3.71, 1.19) | 96 | UMD | Random effect |

(*Continued*)

**Table 1.** (Continued)

| SRMAs | RCTs (N) | Patients (N) | Type of patients | Outcome measurement | Mean difference (95% CI) | I² | Type of mean difference | Analysis model |
|---|---|---|---|---|---|---|---|---|
| Huang, J, 2019 [19] | 3 | 213 | NAFLD/MetS | Change from baseline | **-0.284 (-0.554, -0.013)** | 0 | SMD | Random effect |
| | 4 | 608 | T2DM | Change from baseline | -0.360 (-0.762, 0.043) | 82 | SMD | Random effect |
| Tabrizi, R, 2018 [36] | 5 | 682 | T2DM | Change from baseline | **-0.80 (-1.58, -0.02)** | 95 | SMD | Random effect |
| Zhang, T, 2021 [15] | 4 | 413 | T2DM | Change from baseline | -1.76 (-3.65, 0.13) | 95 | UMD | Random effect |
| **BMI** | | | | | | | | |
| Altobelli, E, 2021 [16] | 3 | 189 | T2DM | Change from baseline | -0.30 (-0.62, 0.02) | 0 | SMD | Random effect |
| Zheng, ZH, 2021 [40] | 3 | 233 | T2DM | Post-intervention value | **-1.75 (-2.23, -1.27)** | 39 | UMD | Fixed effect |
| **TC** | | | | | | | | |
| Altobelli, E, 2021 [16] | 5 | 333 | T2DM | Change from baseline | **-0.30 (-0.53, -0.07)** | 0 | SMD | Random effect |
| Ashtary-Larky, D, 2021 [33] | 1 | 43 | MetS | Change from baseline | 10.4 (-11.09, 31.89) | NR | UMD | Random effect |
| | 2 | 70 | T2DM | Change from baseline | -6.24 (-35.65, 23.17) | 91 | UMD | Random effect |
| Tabrizi, R, 2018 [36] | 8 | 493 | T2DM | Change from baseline | -0.48 (-0.99, 0.02) | 87 | SMD | Random effect |
| Tian, J, 2022 [37] | 9 | 565 | T2DM | Change from baseline | **-8.91 (-14.18, -3.63)** | 29 | UMD | Random effect |
| Yuan, F, 2019 [38] | 14 | 812 | T2DM | Change from baseline | **-12.00 (-20.00, -4.04)** | 46 | UMD | Random effect |
| Zhang, T, 2021 [15] | 4 | 453 | T2DM | Change from baseline | -2.00 (-39.91, 35.91) | 95 | UMD | Random effect |
| Zheng, ZH, 2021 [40] | 5 | 328 | T2DM | Post-intervention value | **-11.30 (-20.69, -1.91)** | 20 | UMD | Fixed effect |
| **TG** | | | | | | | | |
| Altobelli, E, 2021 [16] | 5 | 476 | T2DM | Change from baseline | **-0.57 (-0.83, -0.31)** | 42 | SMD | Random effect |
| Ashtary-Larky, D, 2021 [33] | 3 | 76 | MetS | Change from baseline | -29.11 (-61.92, 3.68) | 40 | UMD | Random effect |
| | 2 | 70 | T2DM | Change from baseline | 16.25 (-51.16, 83.67) | 92 | UMD | Random effect |
| Azhdari, M, 2019 [17] | 5 | 359 | MetS | Change from baseline | **-33.66 (-51.28, -16.04)** | 94 | UMD | Random effect |
| Tabrizi, R, 2018 [36] | 8 | 706 | T2DM | Change from baseline | **-0.85 (-1.63, -0.07)** | 95 | SMD | Random effect |
| Tian, J, 2022 [37] | 9 | 565 | T2DM | Change from baseline | **-18.97 (-36.47, -1.47)** | 81 | UMD | Random effect |
| Yuan, F, 2019 [38] | 12 | 742 | T2DM | Change from baseline | **-24.60 (-48.6, -0.59)** | 84 | UMD | Random effect |
| Zhang, T, 2021 [15] | 4 | 453 | T2DM | Change from baseline | -33.45 (-70.6, 3.71) | 87 | UMD | Random effect |
| Zheng, ZH, 2021 [40] | 4 | 233 | T2DM | Post-intervention value | **-13.99 (-26.91, -1.07)** | 0 | UMD | Fixed effect |
| **LDL-C** | | | | | | | | |
| Altobelli, E, 2021 [16] | 5 | 333 | T2DM | Change from baseline | **-0.28 (-0.52, -0.04)** | 0 | SMD | Random effect |

*(Continued)*

**Table 1.** (Continued)

| SRMAs | RCTs (N) | Patients (N) | Type of patients | Outcome measurement | Mean difference (95% CI) | I² | Type of mean difference | Analysis model |
|---|---|---|---|---|---|---|---|---|
| Ashtary-Larky, D, 2021 [33] | 1 | 43 | MetS | Change from baseline | 16.50 (-9.06, 42.06) | NR | UMD | Random effect |
| | 2 | 70 | T2DM | Change from baseline | -4.72 (-34.37, 24.91) | 94 | UMD | Random effect |
| Tabrizi, R, 2018 [36] | 8 | 493 | T2DM | Change from baseline | -0.10 (-0.78, 0.58) | 93 | SMD | Random effect |
| Tian, J, 2022 [37] | 9 | 565 | T2DM | Change from baseline | -4.01 (-10.96, 2.95) | 50 | UMD | Random effect |
| Yuan, F, 2019 [38] | 13 | 712 | T2DM | Change from baseline | **-10.70 (-18.1, -3.38)** | 51 | UMD | Random effect |
| Zhang, T, 2021 [15] | 5 | 497 | T2DM | Change from baseline | **-6.33 (-12.48, -0.17)** | 89 | UMD | Fixed effect |
| Zheng, ZH, 2021 [40] | 5 | 321 | T2DM | Post-intervention value | **-11.19 (-19.54, -2.84)** | 0 | UMD | Fixed effect |
| **HDL-C** | | | | | | | | |
| Altobelli, E, 2021 [16] | 5 | 333 | T2DM | Change from baseline | 0.22 (-0.08, 0.52) | 46 | SMD | Random effect |
| Ashtary-Larky, D, 2021 [33] | 3 | 76 | MetS | Change from baseline | 5.66 (3.34, 7.98) | 0 | UMD | Random effect |
| | 2 | 70 | T2DM | Change from baseline | 6.84 (-2.85, 16.53) | 93 | UMD | Random effect |
| Azhdari, M., 2019 [17] | 5 | 359 | MetS | Change from baseline | 4.89 (4.59, 5.18) | 99 | UMD | Random effect |
| Tabrizi, R, 2018 [36] | 8 | 493 | T2DM | Change from baseline | 0.36 (-0.19, 0.90) | 89 | SMD | Random effect |
| Tian, J, 2022 [37] | 9 | 565 | T2DM | Change from baseline | 0.32 (-0.74, 1.37) | 19 | UMD | Random effect |
| Yuan, F, 2019 [38] | 16 | 812 | T2DM | Change from baseline | 2.74 (-0.02, 5.50) | 77 | UMD | Random effect |
| Zhang, T, 2021 [15] | 5 | 497 | T2DM | Change from baseline | 2.26 (-2.03, 6.55) | 90 | UMD | Random effect |
| Zheng, ZH, 2021 [40] | 6 | 361 | T2DM | Post-intervention value | 2.92 (1.65, 4.19) | 0 | UMD | Fixed effect |
| **SBP** | | | | | | | | |
| Ashtary-Larky, D, 2021 [33] | 3 | 76 | MetS | Change from baseline | **-11.98 (-21.29, -2.68)** | 69 | UMD | Random effect |
| | 1 | 0 | T2DM | Change from baseline | -3.60 (-8.27, 1.07) | NR | UMD | Random effect |
| Azhdari, M., 2019 [17] | 3 | 280 | MetS | Change from baseline | -1.69 (-4.68, 1.30) | 48 | UMD | Random effect |
| **DBP** | | | | | | | | |
| Azhdari, M, 2019 [17] | 3 | 280 | MetS | Change from baseline | **-2.96 (-5.09, -0.83)** | 49 | UMD | Random effect |
| **CRP** | | | | | | | | |
| Ashtary-Larky, D, 2021 [33] | 2 | 33 | MetS | Change from baseline | -0.64 (-1.52, 0.24) | 86 | UMD | Random effect |
| | 1 | 0 | T2DM | Change from baseline | -3.60 (-8.27, 1.07) | NR | UMD | Random effect |
| Macena, ML, 2022 [35] | 1 | 53 | T2DM | Change from baseline | **-1.60 (-3.14, -0.06)** | NA | SMD | Random effect |
| Panahi, Y, 2015 [20] | 8 | 562 | MetS | Change from baseline | **-2.20 (-3.96, -0.44)** | 95 | UMD | Random effect |

(*Continued*)

**Table 1.** (Continued)

| SRMAs | RCTs (N) | Patients (N) | Type of patients | Outcome measurement | Mean difference (95% CI) | I² | Type of mean difference | Analysis model |
|---|---|---|---|---|---|---|---|---|
| **Hs-CRP** | | | | | | | | |
| Gorabi, AM, 2022 [34] | 3 | 232 | MetS | Change from baseline | -0.81 (-2.72, 1.10) | 94 | SMD | Random effect |
| Gorabi, AM, 2022 [34] | 4 | 279 | T2DM | Change from baseline | -0.17 (-1.75, 1.42) | 66 | SMD | Random effect |

**Abbreviations:** BMI, body mass index; CRP, C-reactive protein; DBP, diastolic blood pressure; FBG, Fasting blood glucose; HbA1C, Hemoglobin A1C; HDL-c, High-density lipoprotein cholesterol; hs-CRP, high-sensitivity C-reactive protein; LDL-c, Low-density lipoprotein cholesterol; MetS, metabolic syndrome; NR, Not reported; pre-DM, pre-diabetic mellitus; SBP, systolic blood pressure; SMD, standardized mean difference; T2DM, type 2 diabetic mellitus; TC, Total cholesterol; TG, Triglyceride; UMD, unstandardized mean difference.

[41,44,50,51,55,58,59,67] RCTs compared CL extract (300–1,950 mg/day), bioavailability-enhanced preparations (80–1,000 mg/day), and whole CL powder (1,000–2,400 mg/day) with a placebo, respectively. One RCT [67] compared a bioavailability-enhanced preparation (1,000 mg/day) with whole CL powder (1,000 mg/day) and placebo. The study duration ranged from 4 to 16 weeks.

In 28 RCTs [20,41–72], the outcomes of interest included FBG (n = 26), HbA1C (n = 22), HOMA-IR (n = 12), insulin (n = 10), TC (n = 22), TG (n = 25), LDL-C (n = 24), HDL-C (n = 24), SBP (n = 15), DBP (n = 15), hs-CRP (n = 10) and uric acid (n = 2) (S6 and S7 Tables in S1 File).

**Risk of bias of included RCTs.** The overall risk of bias in 7 (25%) [43–45,48,49,52,72], 14 (50%) [41,42,46,47,50,51,53–58,60–62,67,68], and 7 (25%) [20,59,63–66,69,70] RCTs were graded as low, some concern, and high, respectively. The most common reasons were an inadequate randomization process and deviation from intended interventions, which were identified by per-protocol analyses (S5 Table in S1 File).

**Primary efficacy outcomes.** Twenty-eight RCTs [20,41–72] involving 2,297 participants and 23 RCTs [20,41,45–51,53–57,59–66,68–72] involving 1,945 participants evaluated FBG and HbA1C levels with the administration of three different CL preparation forms, respectively. Figs 3 and 4 show forest plots of the primary efficacy outcomes. CL supplementation resulted in significant reductions in FBG and HbA1C with UMDs of –8.129 mg/dL (95% CI: –12.175, –4.084 mg/dL; p < 0.001; I² = 75.8%) (Fig 3A) and –0.134% (95% CI: –0.304, –0.037; p < 0.001; I² = 83.0%) (Fig 4A) relative to standard/placebo treatment. In addition, CL significantly reduced FBG and HbA1C changes from the baseline with UMDs of –8.833 mg/dL (95% CI: –13.907, –3.758 mg/dL; p < 0.001; I² = 98.2%) (Fig 3B) and –0.517% (95% CI: –0.707, –0.327; p = 0.004; I² = 61.3%) in HbA1C (Fig 4B).

**Secondary efficacy outcomes.** In comparing the postintervention UMDs, CL supplementation resulted in a significant reduction of –6.199 mg/dL (95% CI: –12.061, –0.336 mg/dL; p = 0.038) for LDL-C and 2.746 mg/dL (95% CI: 0.875, 4.617 mg/dL; p = 0.004) for HDL-C. No significant reductions were observed for TC or TG. In comparing the mean changes from baseline, CL resulted in significant reduction of –12.652 mg/dL (95% CI: –20.066, –5.238 mg/dL; p = 0.001) for TG (S8 Table in S1 File).

CL supplementation also significantly reduced postintervention insulin and DBP levels with UMDs of –0.663 μIU/mL (95% CI: –1.156, –0.171μIU/mL; p = 0.008) and –2.876 mmHg (95%CI: –4.919, –0.833 mmHg; p = 0.006), respectively. In comparing the differences in the mean changes from baseline, significant reductions of 0.444% (95%CI: –0.750, –0.139%; p = 0.004) in HOMA-IR, 0.686 μIU/mL (95%CI: –0.890, –0.481 μIU/mL; p < 0.001) in insulin

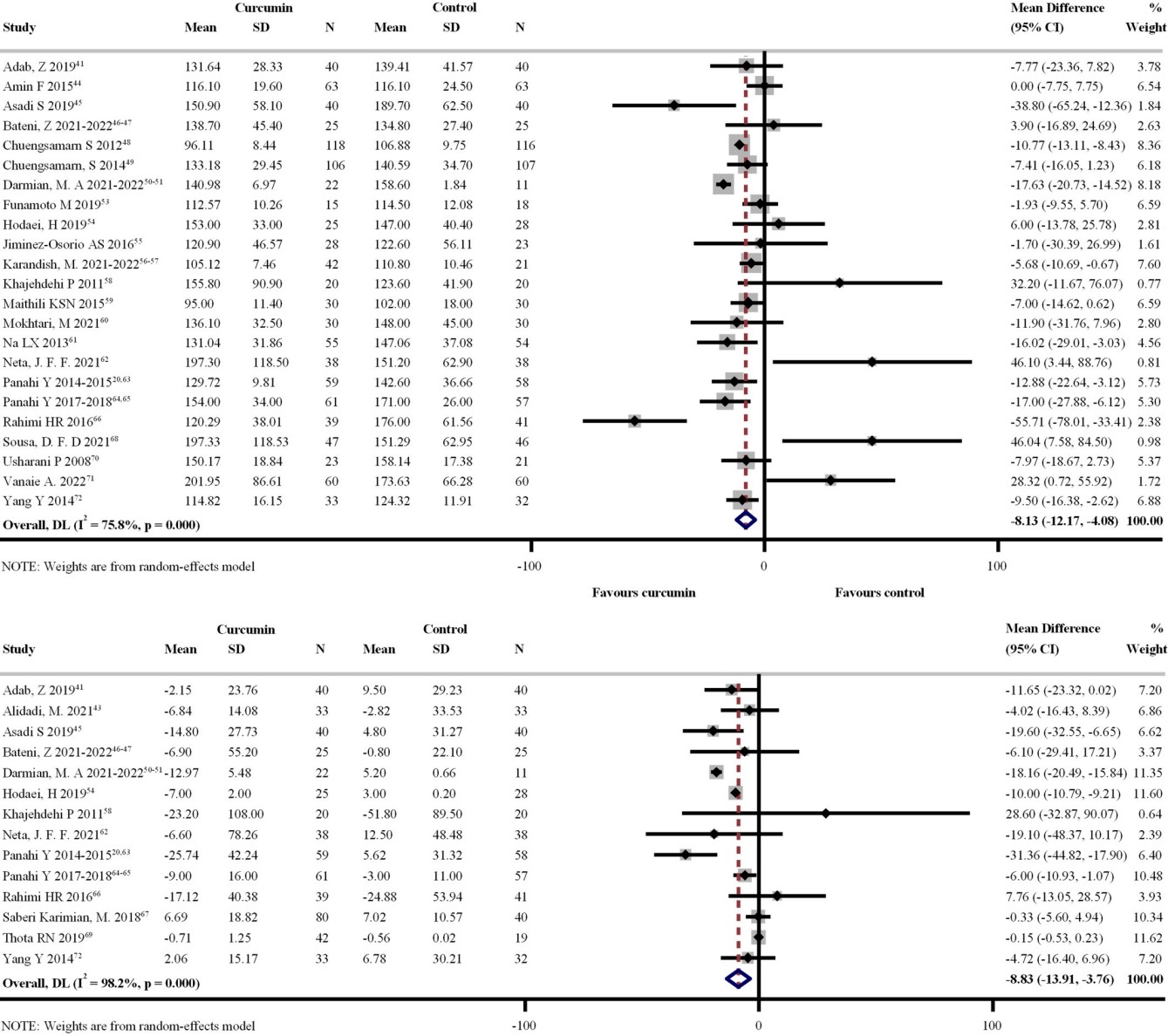

**Fig 3. Forest plot of the differences of fasting blood glucose (mg/dL) within 4 months between *Curcuma longa* (CL) supplementation and control group.** (a) Post-intervention fasting blood glucose value. (b) Change from baseline.

and −0.589 mg/L (−1.158, −0.021 mg/L; p = 0.042) for hs-CRP were observed. All results for the MA are provided in S8 Table in S1 File.

**Subgroup analyses.** Figs 5 and 6 show forest plots of the subgroup analyses. For the different CL preparation forms, a significant reduction in FBG from baseline was observed for CL extract and bioavailability-enhanced preparations. However, a significant reduction in HbA1C was noted for all CL preparation forms. In addition, a subgroup analysis based on dose showed that the reduction in FBG and HbA1C levels increased with higher doses. Subgroup analysis based on types of patients showed a significant reduction in FBG from baseline in T2DM and MetS, whereas a significant reduction in HbA1C was observed only in T2DM. Furthermore, the findings of the subgroup analysis based on patient baseline characteristics were consistent

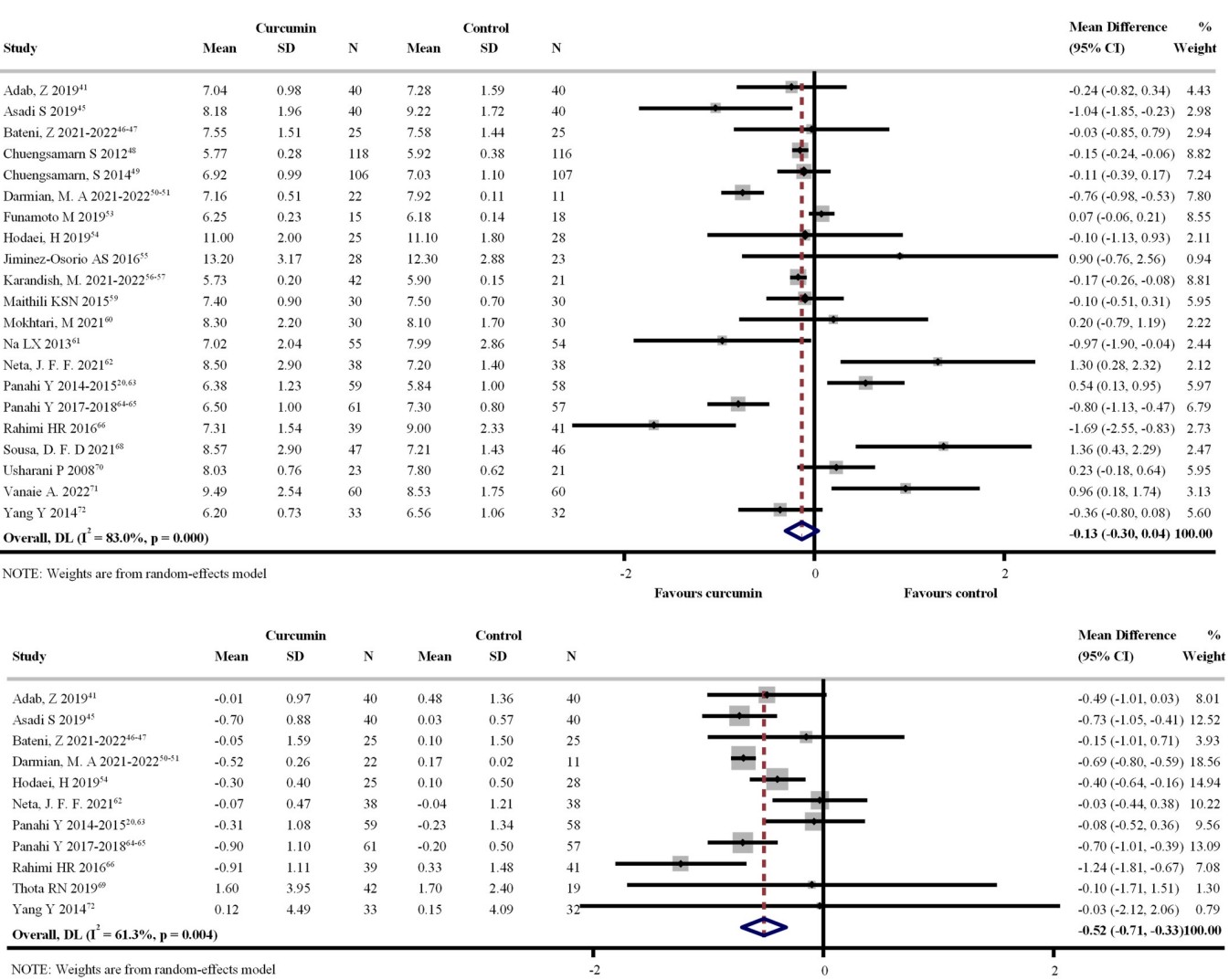

**Fig 4. Forest plot of the differences of hemoglobin A1C (%) within 4 months between *Curcuma longa* (CL) supplementation and control group.** (a) Post-intervention HbA1C value. (b) Change from baseline.

with those of the main analysis. Details of the subgroup analyses of the primary efficacy outcomes are presented in S9 and S10 Tables in S1 File. The details of the post-hoc analysis of the secondary outcomes stratified by CL preparation forms are provided in S11 Table in S1 File.

**Sensitivity analyses and publication bias.** Three sensitivity analyses were conducted: 1) analysis using the fixed effects model; 2) exclusion of trials with a high risk of bias; and 3) exclusion of small-sized studies ($<25^{th}$ percentile). The results of these analyses remained consistent with those of the overall analysis (S12 Table in S1 File). Publication bias was assessed using funnel plots and Egger's test for all outcomes. The funnel plots were symmetrical (S30–S42 Figs in S1 File) corresponding to the results of the Egger's tests, suggesting no publication bias (S13 Table in S1 File), except for postintervention FBG and BMI outcomes.

**Quality of evidence.** The quality of direct evidence for all primary outcomes was generally rated moderate. The grade was decreased by inconsistency due to a large heterogeneity (large $I^2$). More details on the quality of evidence are provided in S14 Table in S1 File.

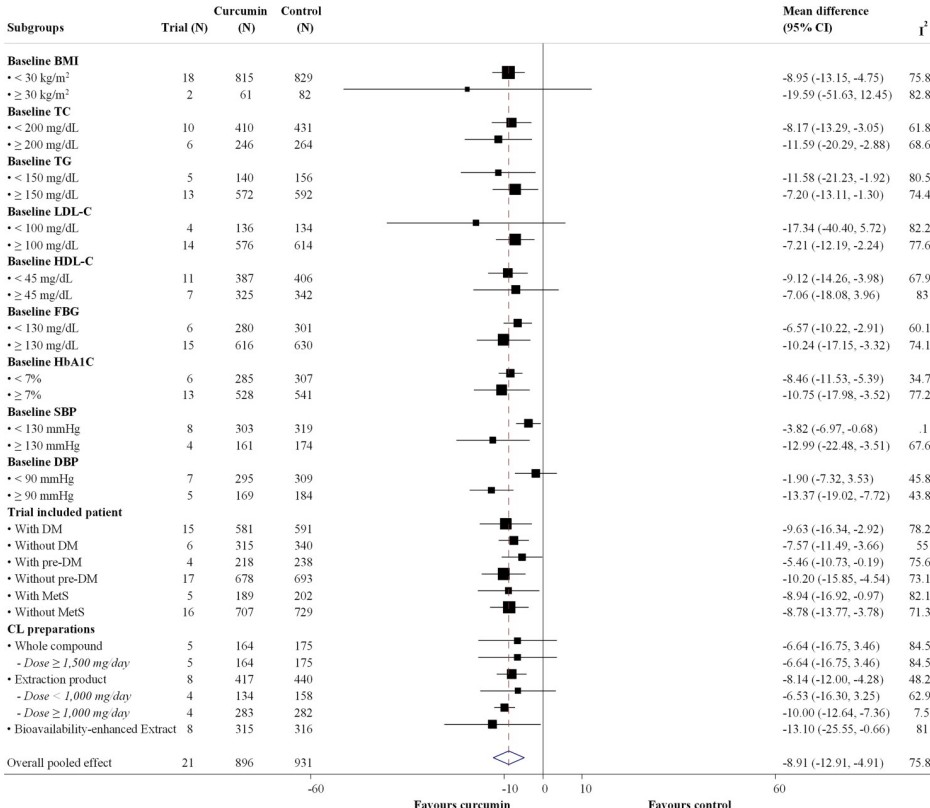

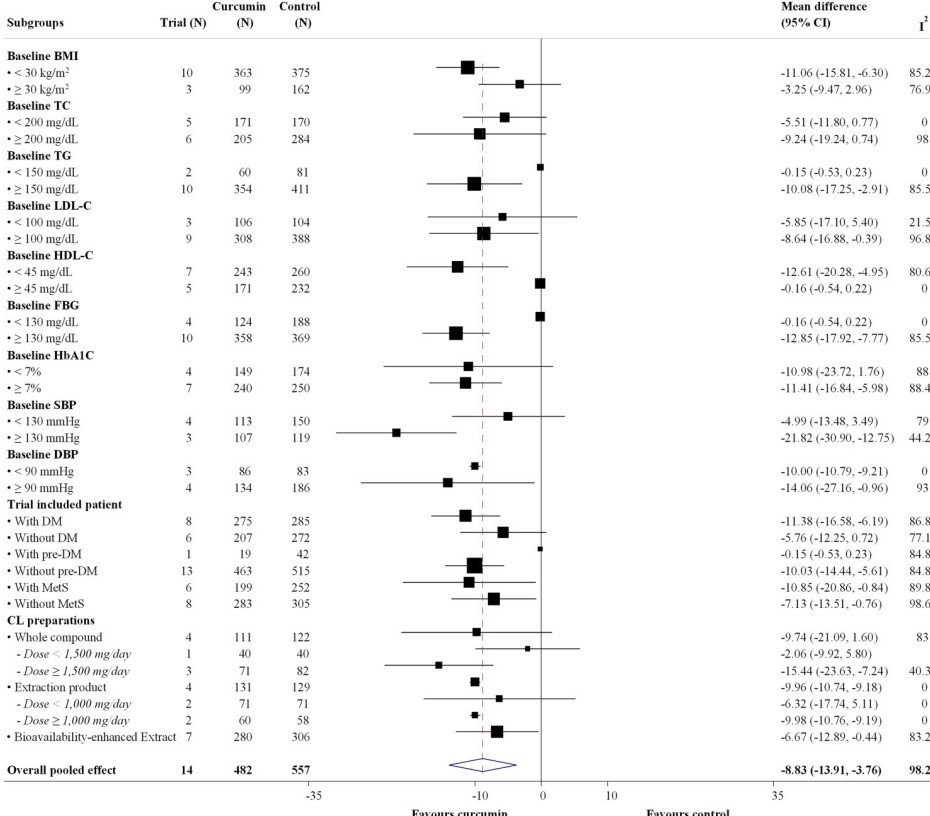

**Fig 5. Subgroup analysis of difference of fasting blood glucose (mg/dL) within 4 months between *Curcuma longa* (CL) supplementation and control group.** (a) Post-intervention values. (b) Change from baseline. Abbreviations: BMI, body mass index; CL, Curcuma longa; DBP, diastolic blood pressure; DM, diabetic mellitus; FBG, fasting blood glucose; HbA1C, hemoglobin A1C; HDL-c, high-density lipoprotein cholesterol; LDL-c, low-density lipoprotein cholesterol; MetS, metabolic syndrome; NA, not applicable; pre-DM, pre-diabetic mellitus; SBP, systolic blood pressure; TC, total cholesterol; TG, triglyceride.

## Discussion

Our UR that include 14 SRMAs and an updated MA of RCTs indicated significant reductions due to CL supplementation in FBG and HbA1C of about 8 mg/dL and 0.134% to 0.517%, respectively. In addition, CL extract and bioavailability-enhanced preparations could significantly reduce both FBG and HbA1C.

CL has long been used in Eastern medicine for treating various diseases and symptoms and is one of the most studied medicinal plants on earth. The most recent scientific evidence suggests that CL exhibits various cellular actions that can mitigate diabetes [14]. *In vitro* studies have shown that CL increases insulin expression and secretion by activating the phosphatidylinositol-3-kinase/protein kinase B/glucose transporter 2 (PI3K/Akt/GLUT2) signaling pathway and upregulates insulin mRNA expression [73]. In mouse models, CL attenuated oxidative stress, dose-dependently reducing the apoptosis of pancreatic β-cells [74]. In hepatocytes and adipocytes, CL reduced glucose uptake by inhibiting the translocation of GLUT4 from the cytosol to the plasma membrane via the insulin receptor substance 1/PI3K/Akt signaling pathway [75]. In the gut, CL inhibited the enzymes involved in glucose digestion, i.e., α-amylase and α-glucosidase [76]. CL also stimulated the secretion of glucagon-like peptide-1, and incretins, which are a validated pathway of important classes of anti-diabetic drugs [77]. Notably, CL upregulated peroxisome proliferator-activated receptor-γ (PPAR-γ) expression in liver cells, thereby downregulating p65 nuclear factor kappa B, a protein complex central to the inflammation [78]. CL curcuminoids also inhibited advanced glycation end-products which are the key mediators associated with metabolic complications [76].

Three forms of CL preparation are most commonly used, namely, whole CL, CL extract and bioavailability-enhanced preparations. The different preparation forms may be administered with different types and doses of bioactive compounds [79]. Studies have shown that bioavailability-enhanced preparations, such as curcumin-phosphatidylcholine phytosome complex, curcumin micelles, water-soluble curcumin formulation containing, or colloidal nanoparticle formulation, increased the absorption of curcumin by 6.9–20.0 times compared with whole CL preparation [79]. Therefore, an assessment of the effects of different CL preparations on outcomes is imperative.

Results from both the UR and the updated MA suggest that CL preparations can reduce FBG and modest HbA1C. Interestingly, subgroup analysis showed that CL exhibited beneficial effects on glycemic management, especially in T2DM, pre-diabetes, MetS, and high SBP/DBP. Similar positive effect on glucose homeostasis had also been proved in nonalcoholic fatty liver disease [80]. Regardless of the preparations used, a dose-response relationship was observed, where higher doses of the same preparation elicited stronger effects. In addition, significant reductions in other metabolic parameters were observed, further confirming the potent effects of CL that may be expressed via multiple pathways.

### Limitations and strengths

Our study has several limitations. First, the quality of SRMAs were all considered critically low. Second, some relevant outcomes (i.e., HOMA-IR, insulin, blood pressure, CRP) were

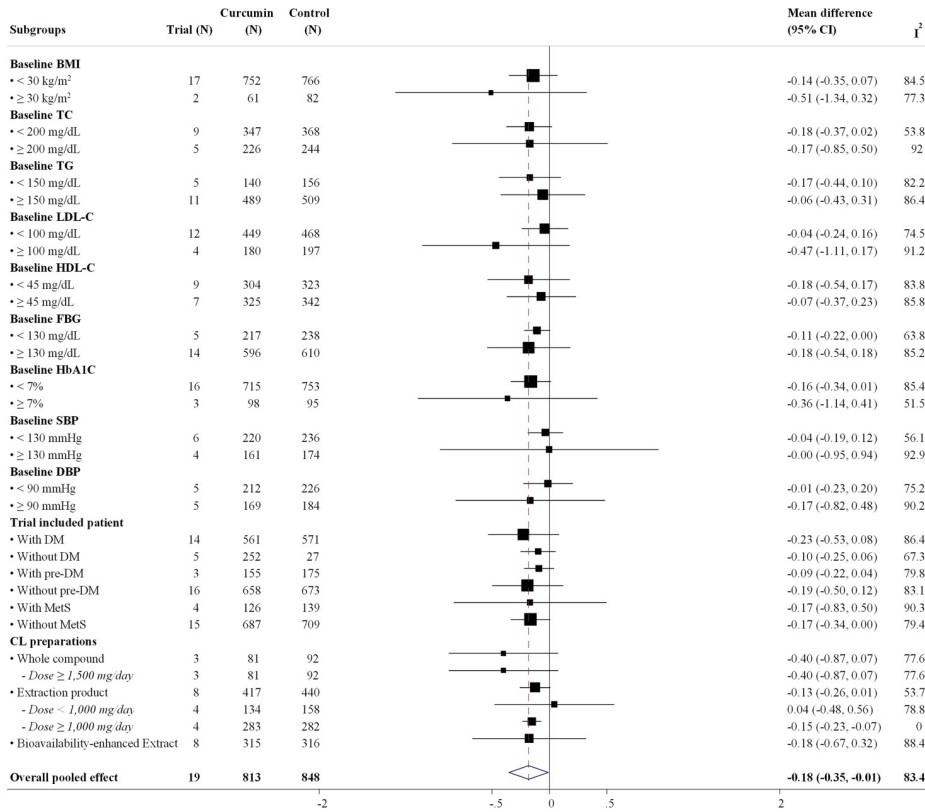

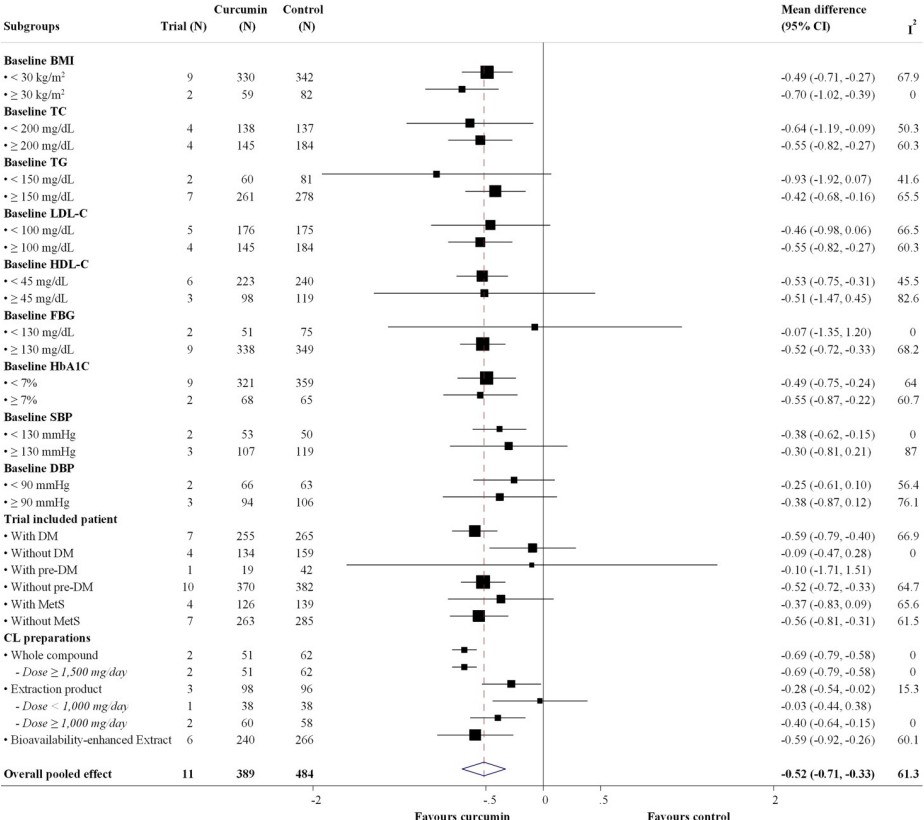

**Fig 6. Forest plot of subgroup analysis of difference in change of hemoglobin A1C (%) within 4 months between *Curcuma longa* (CL) supplementation and control group.** (a) Post-intervention values. (b) Change from baseline. Abbreviations: BMI, body mass index; CL, Curcuma longa; DBP, diastolic blood pressure; DM, diabetic mellitus; FBG, fasting blood glucose; HbA1C, hemoglobin A1C; HDL-c, high-density lipoprotein cholesterol; LDL-c, low-density lipoprotein cholesterol; MetS, metabolic syndrome; NA, not applicable; pre-DM, pre-diabetic mellitus; SBP, systolic blood pressure; TC, total cholesterol; TG, triglyceride.

reported in only a few studies. Hence, the effect sizes were not accurately estimated. Third, differences in bioactive compounds contained in various preparations may exist, which may affect the clinical effects. Such differences are due to varying qualities of the raw materials, extraction methods, and formulation techniques.

## Conclusions

Our review revealed that CL supplementation can significantly reduce FBG and HbA1C levels in T2DM and mitigate related conditions, including prediabetes and MetS, after short-term use. The effects appear consistent regardless of the form of CL preparation used. A dose-response relationship is suggested in the findings, where higher doses of the same preparation elicited stronger responses. In addition, significant reductions in other metabolic parameters were observed. Further studies should be conducted to assess whether the effects on glycemic management can be sustained on a long-term basis and/or whether they can reduce the risk of diabetic complications.

## Supporting information

**S1 Checklist. PRISMA 2020 checklist.**
(DOCX)

**S1 File.**
(ZIP)

## Author Contributions

**Conceptualization:** Surakit Nathisuwan, Ammarin Thakkinstian.

**Formal analysis:** Peerawat Jinatongthai, Wipharak Rattanavipanon.

**Investigation:** Thanika Pathomwichaiwat, Peerawat Jinatongthai, Napattaoon Prommasut, Kanyarat Ampornwong, Wipharak Rattanavipanon.

**Methodology:** Ammarin Thakkinstian.

**Supervision:** Surakit Nathisuwan, Ammarin Thakkinstian.

**Validation:** Thanika Pathomwichaiwat, Peerawat Jinatongthai.

**Visualization:** Peerawat Jinatongthai.

**Writing – original draft:** Thanika Pathomwichaiwat, Peerawat Jinatongthai, Surakit Nathisuwan, Ammarin Thakkinstian.

**Writing – review & editing:** Wipharak Rattanavipanon, Surakit Nathisuwan, Ammarin Thakkinstian.

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
