## [Decision Letter · Decision Letter 0]

11 Jan 2023

PONE-D-22-31206Effects of Curcuma longa supplementation on glucose metabolism in diabetes mellitus and metabolic syndrome: an umbrella review and updated meta-analysisPLOS ONE

Dear Dr. Jinatongthai,

Thank you for submitting your manuscript to PLOS ONE. After careful consideration, we feel that it has merit but does not fully meet PLOS ONE’s publication criteria as it currently stands. Therefore, we invite you to submit a revised version of the manuscript that addresses the points raised during the review process.

We look forward to receiving your revised manuscript.

Kind regards,

Mohamed Ezzat Abd El-Hack

Academic Editor

PLOS ONE

Journal Requirements:

Reviewers' comments:

Reviewer's Responses to Questions

**Comments to the Author**

1. Is the manuscript technically sound, and do the data support the conclusions?

Reviewer #1: Yes

2. Has the statistical analysis been performed appropriately and rigorously? 

Reviewer #1: Yes

3. Have the authors made all data underlying the findings in their manuscript fully available?

Reviewer #1: No

4. Is the manuscript presented in an intelligible fashion and written in standard English?

Reviewer #1: Yes

5. Review Comments to the Author

Reviewer #1: This is a well designed, well written manuscript. Only, I recommend to discuss all related papers such as https://doi.org/10.1038/s41430-018-0382-9. Although some papers are on other disorders, they have measured similar blood values.

6. PLOS authors have the option to publish the peer review history of their article (what does this mean?). If published, this will include your full peer review and any attached files.

Reviewer #1: No

---

## [Author Response · Author response to Decision Letter 0]

28 Jan 2023

Dear editor,

Thank you for your valuable recommendation and comments. We have checked and revised according to your comments. The revised manuscript and point-by-point responses to the additional requirements and reviewer’s comments files have been attached with this submission. We look forward to hearing from you soon.

Sincerely,

---

## [Decision Letter · Decision Letter 1]

24 May 2023

PONE-D-22-31206R1

Effects of Curcuma longa supplementation on glucose metabolism in diabetes mellitus and metabolic syndrome: an umbrella review and updated meta-analysis

PLOS ONE

Dear Dr. Jinatongthai,

Thank you for submitting your manuscript to PLOS ONE. After careful consideration, we feel that it has merit but does not fully meet PLOS ONE’s publication criteria as it currently stands. Therefore, we invite you to submit a revised version of the manuscript that addresses the points raised during the review process.

We look forward to receiving your revised manuscript.

Kind regards,

Mohammad Reza Mahmoodi, Ph.D.

Academic Editor

PLOS ONE

 Journal Requirements:

**Additional Editor Comments:**

**# Please make the minor editorial changes in wording suggested by the reviewer.  After that, I am happy to accept the manuscript for publication.**

**# To ensure the Editor and Reviewers will be able to recommend that your revised manuscript is accepted, please pay careful attention to each of the comments that have been pasted underneath this email. This way we can avoid future rounds of clarifications and revisions, moving swiftly to a decision.**

**# Please highlight all the corrections and changes made based on the second peer reviewer's suggestions. Then send one highlighted version and one without highlighting for review.**

Reviewers' comments:

Reviewer's Responses to Questions

**Comments to the Author**

1. If the authors have adequately addressed your comments raised in a previous round of review and you feel that this manuscript is now acceptable for publication, you may indicate that here to bypass the “Comments to the Author” section, enter your conflict of interest statement in the “Confidential to Editor” section, and submit your "Accept" recommendation.

Reviewer #1: All comments have been addressed

Reviewer #2: (No Response)

2. Is the manuscript technically sound, and do the data support the conclusions?

Reviewer #1: Yes

Reviewer #2: Yes

3. Has the statistical analysis been performed appropriately and rigorously? 

Reviewer #1: Yes

Reviewer #2: Yes

4. Have the authors made all data underlying the findings in their manuscript fully available?

Reviewer #1: (No Response)

Reviewer #2: Yes

5. Is the manuscript presented in an intelligible fashion and written in standard English?

Reviewer #1: Yes

Reviewer #2: Yes

6. Review Comments to the Author

Reviewer #1: All my comments are addressed. The manuscript can be accepted in its current format.

there is no added comments.

Reviewer #2: The current manuscript was an umbrella review that was designed and written well. The only minor flaw in this manuscript is the lack of an expansion discussion regarding the reasons and mechanisms of the effect of turmeric on reducing glycemic indices. Curcuma longa is a food additive most studied in the scientific domain.

Title:

1. Title change to Effects of turmeric (Curcuma longa) supplementation on glucose metabolism in diabetes mellitus and metabolic syndrome: an umbrella review and updated meta-analysis

Abstract:

2. Line 20: Curcuma longa convert to turmeric (Curcuma longa)

3. Lines 20 & 21: “its related compounds on glycemic” convert to “its related bioactive compounds on glycemic”

4. Line 22 and the remainder of abstract: CL convert to turmeric

5. Lines 22 & 23: “different CL preparation-forms” convert to “different CL preparation forms” OR “different CL preparations” (Conversion should be accomplished throughout the text of the manuscript)

6. Line 26 & 27: As the authors know, Scopus and Web of Science are big databases. Why did the authors not use these databases? The number of articles on a topic may be 3-4 times more in Scopus database than PubMed database.

Introduction:

7. Line 73. CL preparation forms convert to turmeric (CL) preparation forms

8. Line 79. “the effects of curcumin and its related compounds on glycemic” converted to “the effects of turmeric preparations and its related bioactive compounds on glycemic”

Methods:

9. In flowchart (Fig 1), the authors identified that 1750 abstracts were extracted from Scopus database. Why this database didn’t mention in the abstract together the other databases?

Types of interventions

10. Line 115: “curcuminoid standardized extract” convert to “standardized curcuminoid extract”

11. Please kindly correct S1 Table to Table S1, S2 Table to Table S2, etc. throughout the manuscript text.

Outcomes of interest

12. According to NCEP ATP III, MetS was identified based on five criteria. Among five criteria, there aren’t serum uric acid and hsCRP. The authors describe, why these biomarkers selected for MetS outcomes.

Data synthesis and statistical analysis

13. Line 161: “lipid profiles” modify to “lipoprotein profiles”

Results:

14. As I noted in comment 11, all “S1, 2, 3, ……. Table” modify to “Table S1, 2, 3, …….” In the manuscript text.

15. The authors should be modified “S1, 2, 3, …… Fig” to “Fig S1, 2, 3, ……” throughout in manuscript text similar to Tables.

Table 1:

16. BMI is abbreviate body mass index not bone mass index. Modify it in footnote of Table 1.

17. Would you please review Huang [19] study rows for biomarkers in table 1? Is there any inconsistency in recorded data in table 1?

Figs:

18. Did the professor Ammarin Thakkinstian perform all the meta-analyses of this study?

19. Line 321: “bone mass index” modify to “body mass index” in Abbreviation of Fig 6.

20. Line 332: “S30–S42 Figs” modify to “Figs S30-S42”. The other Tables and Figs should be corrected.

21. Lines 367-368: What are “bioavailability-enhanced preparations” in the other studies? Modify to “bioavailability-enhanced preparations such as ………….”

22. The last paragraph of the discussion section should be separated under “Limitations and Strengths”.

23. As the authors converted Type 2 diabetes mellitus to T2DM, metabolic syndrome modify to MetS in throughout the manuscript text.

24. Which of the tables or figures in the supporting information section are more important to appear in the manuscript? Or to remain in the supporting information section?

Congratulations

7. PLOS authors have the option to publish the peer review history of their article (what does this mean?). If published, this will include your full peer review and any attached files.

Reviewer #1: No

Reviewer #2: No

---

## [Author Response · Author response to Decision Letter 1]

6 Jun 2023

Comment #1 Title change to Effects of turmeric (Curcuma longa) supplementation on glucose metabolism in diabetes mellitus and metabolic syndrome: an umbrella review and updated meta-analysis. 

Response #1 The title was changed as the reviewer recommended.

Lines 1–2, 

Effects of turmeric (Curcuma longa) supplementation on glucose metabolism in diabetes mellitus and metabolic syndrome: an umbrella review and updated meta-analysis

Comment #2 Line 20: Curcuma longa convert to turmeric (Curcuma longa) 

Response #2 The common name was added as recommended.

Line 20,

“…(RCTs) of turmeric (Curcuma longa, CL) and its related bioactive compounds…”

Comment #3 Lines 20 & 21: "its related compounds on glycemic" convert to "its related bioactive compounds on glycemic" 

Response #3 The sentence was changed as the reviewer recommended.

Line 20–21,

“…and its related bioactive compounds on glycemic…”

Comment #4 Line 22 and the remainder of abstract: CL convert to turmeric

Response #4 We appreciate your advice. We acknowledge turmeric is a well-known common name of Curcuma longa L. However, using the turmeric term could mislead many plants, such as, C. inodora Blatt., C. pseudomontana J.Graham, C. caulina J.Graham, C. roscoeana Wall., and Zingiber officinale Roscoe [1]. Therefore, to prevent common taxonomical errors [2], we agree to preserve Curcuma longa and its abbreviation, CL, according to the original manuscript.

References:

1. POWO (2023). "Plants of the World Online. Facilitated by the Royal Botanic Gardens, Kew. Published on the Internet; https://powo.science.kew.org/results?q=turmeric. Retrieved 30 May 2023.

2. Bennett, B. C., & Balick, M. J. (2014). Does the name really matter? The importance of botanical nomenclature and plant taxonomy in biomedical research. Journal of Ethnopharmacology, 152(3), 387–392.

Comment #5 Lines 22 & 23: "different CL preparation-forms" convert to "different CL preparation forms" OR "different CL preparations" (Conversion should be accomplished throughout the text of the manuscript) 

Response #5 The words have changed as reviewer recommended from “CL preparation-forms” to “CL preparation forms” and checked throughout the manuscript.

Comment #6 Line 26 & 27: As the authors know, Scopus and Web of Science are big databases. Why did the authors not use these databases? The number of articles on a topic may be 3-4 times more in Scopus database than PubMed database. 

Response #6 We apologize for the typographical error. Scopus was one of the included databases according to our protocol. The abstract was revised and corrected.

Lines 26–27,

“…The MEDLINE, Embase, The Cochrane Central Register of Control Trials, and Scopus databases were searched…”

Comment #7 Line 73. CL preparation forms convert to turmeric (CL) preparation forms

Response #7 Please refer to the response to reviewer comments #4.

Comment #8 Line 79. "the effects of curcumin and its related compounds on glycemic" converted to "the effects of turmeric preparations and its related bioactive compounds on glycemic"

Response #8 Line 79,

“…curcumin and its related bioactive compounds on glycemic…”

Comment #9 In flowchart (Fig 1), the authors identified that 1750 abstracts were extracted from Scopus database. Why this database didn't mention in the abstract together the other databases?

Response #9 The Scopus database was included in the systematic searching according to our study protocol. We followed the advice by correcting the abstract.

Line 26-27,

“…The MEDLINE, Embase, The Cochrane Central Register of Control Trials, and Scopus databases…”

Comment #10 Line 115: "curcuminoid standardized extract" convert to "standardized curcuminoid extract"

Response #10 Line 115,

“…e.g., a standardized curcuminoid extract; and 3)…”

Comment #11 Please kindly correct S1 Table to Table S1, S2 Table to Table S2, etc. throughout the manuscript text.

Response #11 We followed the advice of the reviewer’s comments by correcting the label of supplementary table throughout the manuscript.

Comment #12 According to NCEP ATP 111, Mets was identified based on five criteria. Among five criteria, there aren't serum uric acid and hsCRP. The authors describe, why these biomarkers selected for Mets outcomes.

Response #12 We followed our study protocol. Although the exact mechanism has not yet been clarified., several evidence implied that an elevation of serum uric, C-reactive protein (CRP) and high sensitivity C-reactive protein (hs-CRP) are associated with an increased risk of developing metabolic syndrome components, such as hypertension, insulin resistance, and dyslipidemia [1-3]. With a certain number of published evidence and none of the previous SRMAs have been investigated related to these markers, therefore we included them as one of outcomes of interest to provide the most up-to-date and comprehensive evidence. We added more details to acknowledge this comment as below.

References:

1. López-Olivo MA, et al. Serum uric acid as a predictor of incident metabolic syndrome: a secondary analysis of a longitudinal cohort study. Arthritis Care Res. 2013;65(12):2026-2031.

2. Kanbay M, et al. Uric acid and metabolic syndrome: lessons from a large, multicenter study. PLoS ONE. 2018;13(3):e0194125.

3. Lin SD, et al. Hyperuricemia and metabolic syndrome: associations with endothelial dysfunction. Am J Med Sci. 2010;339(3):233-239.

 Data synthesis and statistical analysis

Comment #13 Line 161: "lipid profiles" modify to "lipoprotein profiles"

Response #13 We followed our study protocol. A lipid profile typically includes measurements of total cholesterol, high-density lipoprotein cholesterol (HDL-C), low-density lipoprotein cholesterol (LDL-C), and triglycerides,which are the common markers used in a routine clinical practice and also general clinical research. Whereas the term “lipoprotein profiles” may refer to subclasses of lipoproteins, such as, LDL particles (including small, dense LDL), very low-density lipoprotein (VLDL) particles, and HDL subclasses. To provide the clinical aspects for the readers, we agreed to preserve the term “lipid profiles” according to the original manuscript.

Comment #14 As I noted in comment 11, all "S1, 2, 3, ....... Table" modify to "Table S1, 2, 3, ....... " In the manuscript text

Response #14 The words were changed as reviewer recommended and checked throughout the manuscript.

Comment #15 The authors should be modified "S1, 2, 3, ...... Fig" to "Fig S1, 2, 3, ...... " throughout in manuscript text similar to Tables.

Response #15 The words were changed as reviewer recommended and checked throughout the manuscript.

Comment #16 BMI is abbreviate body mass index not bone mass index. Modify it in footnote of Table 1. 

Response #16 We apologize for the typographical error. The word was corrected and checked throughout the manuscript.

Lines 223 and 310, 

“Abbreviations: BMI, body mass index;…”

Comment #17 Would you please review Huang [19] study rows for biomarkers in table 1? Is there any inconsistency in recorded data in table 1?

Response #17 We have rechecked all the records. All reported data is correct. The number of reported outcomes may difference within its original study because of the difference in number of subpopulations reported for each outcome.

Comment #18 Did the professor Ammarin Thakkinstian perform all the meta-analyses of this study?

Response #18 We added more details about the author’s responsibility for “Data synthesis and statistical analysis” as described below. We also corrected the method on STATA program and the citation error as below.

Lines 162-166, 

“All analyses were performed by P.J., T.P., and W.R. using STATA® version 17.0 (StataCorp, College Station, Texas, USA) along with the self-programmed STATA® for meta-analysis described elsewhere[32]. A.T. and S.N. provided technical analysis advice. A p-value ≤ 0.05 was considered statistically significant.”

Line 513

32. Palmer TM, Sterne JAC. Meta-analysis in Stata: An Updated Collection from the Stata Journal: Stata Press; 2016.

Comment #19 Line 321: "bone mass index" modify to "body mass index" in Abbreviation of Fig 6.

Response #19 We apologize for the typographical error. The word was corrected.

Line 318,

“Abbreviations: BMI, body mass index;…”

Comment #20 Line 332: "S30-S42 Figs" modify to "Figs S30-S42". The other Tables and Figs should be corrected.

Response #20 The words were changed as reviewer recommended and checked throughout the manuscript.

Comment #21 Lines 367-368: What are "bioavailability-enhanced preparations" in the other studies? Modify to "bioavailability-enhanced preparations such as 

Response #21 We followed the advice of the reviewer’s comments by adding the details to explain about the preparation.

Lines 365–367,

“…bioavailability-enhanced preparations, such as curcumin-phosphatidylcholine phytosome

complex, curcumin micelles, water-soluble curcumin formulation containing, or colloidal nanoparticle formulation, increased the absorption…”

Comment #22 The last paragraph of the discussion section should be separated under "Limitations and Strengths".

Response #22 The last paragraph was separated as recommended.

Line 378,

“Limitations and Strengths

Our study has several limitations. First,…”

Comment #23 As the authors converted Type 2 diabetes mellitus to T2DM, metabolic syndrome modify to Mets in throughout the manuscript text.

Response #23 The metabolic syndrome was converted to MetS throughout the manuscript.

Comment #24 Which of the tables or figures in the supporting information section are more important to appear in the manuscript? Or to remain in the supporting information section?

Response #24 We have reviewed throughout the manuscript. All major results with graphic have been reported in main manuscript.

---

## [Decision Letter · Decision Letter 2]

15 Jun 2023

PONE-D-22-31206R2

Effects of turmeric (Curcuma longa) supplementation on glucose metabolism in diabetes mellitus and metabolic syndrome: an umbrella review and updated meta-analysis.

PLOS ONE

Dear Dr. Jinatongthai,

Thank you for submitting your manuscript to PLOS ONE. After careful consideration, we feel that it has merit but does not fully meet PLOS ONE’s publication criteria as it currently stands. Therefore, we invite you to submit a revised version of the manuscript that addresses the points raised during the review process.

As the authors know, the publication a manuscript depends on satisfaction of the reviewers of a manuscript. I am sure that these minor changes will also be applied in the manuscript and the dear authors' article will reach the publication process. 

We look forward to receiving your revised manuscript.

Kind regards,

Mohammad Reza Mahmoodi, Ph.D.

Academic Editor

PLOS ONE

Journal Requirements:

Reviewers' comments:

Reviewer's Responses to Questions

**Comments to the Author**

1. If the authors have adequately addressed your comments raised in a previous round of review and you feel that this manuscript is now acceptable for publication, you may indicate that here to bypass the “Comments to the Author” section, enter your conflict of interest statement in the “Confidential to Editor” section, and submit your "Accept" recommendation.

Reviewer #2: All comments have been addressed

2. Is the manuscript technically sound, and do the data support the conclusions?

Reviewer #2: Yes

3. Has the statistical analysis been performed appropriately and rigorously? 

Reviewer #2: Yes

4. Have the authors made all data underlying the findings in their manuscript fully available?

Reviewer #2: Yes

5. Is the manuscript presented in an intelligible fashion and written in standard English?

Reviewer #2: Yes

6. Review Comments to the Author

Reviewer #2: Some comments were not accomplished. The authors should be modified the manuscript based on the following comments that did not modified.

8. Line 79. “the effects of curcumin and its related compounds on glycemic” converted to “the effects of turmeric preparations and its related bioactive compounds on glycemic”

12. “Outcomes related to MetS were also collected …” should be modified to “The other selected outcomes were also collected …”

Based on the response to comment 21, comment 12 should be modified! On the other hand, unfortunately, the authors did not understand my comment (number 12). Some biomarkers were measured for participants took part in a study either might exaggerated some main biomarkers and patients outcome Or they are measured according to the wishes of the authors. Therefore, authors should be subject to the reviewer's comments. If the authors want measure association metabolic syndrome with C-peptide; we can assume that this indicator is one of the main indicators of metabolic syndrome? Such an assumption is completely wrong. Researchers can measure the correlation/association/relationship of a biomarker or an outcome in a disease based on a strong implementation necessity, but they cannot relate that biomarker as a main indicator to that disease.

13. The authors mentioned that they followed the study protocol. This explanation was rejected. If a phrase or phrases was/were wrong, we could not compliance with it/them. Hence, the phrases should be modified to “lipid and lipoprotein profile”.

17. I am not convinced.

19. Bone mass index (BMI) was not a typographical error. It was a wrong compliance.

24. The authors didn't understand what I meant. I mean, which of the tables and figures in the appendix section should be added to the text of the main manuscript?

7. PLOS authors have the option to publish the peer review history of their article (what does this mean?). If published, this will include your full peer review and any attached files.

Reviewer #2: No

---

## [Author Response · Author response to Decision Letter 2]

4 Jul 2023

#8 “the effects of curcumin and its related compounds on glycemic” converted to “the effects of turmeric preparations and its related bioactive compounds on glycemic” 

We followed the advice of the reviewer’s comments by modifying the following sentences.

Line 79,

“…the effects of turmeric preparations and its related bioactive compounds on glycemic…”

#12 “Outcomes related to MetS were also collected …” should be modified to “The other selected outcomes were also collected …”

Based on the response to comment 21, comment 12 should be modified! On the other hand, unfortunately, the authors did not understand my comment (number 12). Some biomarkers were measured for participants took part in a study either might exaggerated some main biomarkers and patients outcome Or they are measured according to the wishes of the authors. Therefore, authors should be subject to the reviewer's comments. If the authors want measure association metabolic syndrome with C-peptide; we can assume that this indicator is one of the main indicators of metabolic syndrome? Such an assumption is completely wrong. Researchers can measure the correlation/association/relationship of a biomarker or an outcome in a disease based on a strong implementation necessity, but they cannot relate that biomarker as a main indicator to that disease. 

We followed the advice of the reviewer’s comments by modifying the following sentences.

Line 126,

“…The other selected outcomes were also collected including…”

#13 The authors mentioned that they followed the study protocol. This explanation was rejected. If a phrase or phrases was/were wrong, we could not compliance with it/them. Hence, the phrases should be modified to “lipid and lipoprotein profile”. 

We followed the advice of the reviewer’s comments by modifying the following sentences.

Line 160,

“…baseline characteristics of patients (including mean body mass index [BMI], lipid and lipoprotein profiles, blood pressure…”

#17 I am not convinced.

Would you please review Huang [19] study rows for biomarkers in table 1? Is there any inconsistency in recorded data in table 1?

We have gone through this paper and have changed numbers to be consistent with the original paper. Please accept our sincere apology for our mistake.

#19 Bone mass index (BMI) was not a typographical error. It was a wrong compliance.

We apologize for our inaccurate explanation of our response. It was a wrong compliance. The manuscript has been corrected according to the reviewer’s comments.

#24 The authors didn't understand what I meant. I mean, which of the tables and figures in the appendix section should be added to the text of the main manuscript?

After careful consideration, we believe that the information included in the main manuscript is sufficient to maintain concise and focused data according to the objective of study. We also cited in the main manuscript to refer related additional information to the supplementary.

In case you feel that we still do not fully understand your question well enough, we kindly request you to clearly specify those details. We will thoroughly review your suggestions and make necessary revisions accordingly. Thank you.

---

## [Decision Letter · Decision Letter 3]

10 Jul 2023

Effects of turmeric (Curcuma longa) supplementation on glucose metabolism in diabetes mellitus and metabolic syndrome: an umbrella review and updated meta-analysis.

PONE-D-22-31206R3

Dear Dr. Jinatongthai,

We’re pleased to inform you that your manuscript has been judged scientifically suitable for publication and will be formally accepted for publication once it meets all outstanding technical requirements.

Kind regards,

Mohammad Reza Mahmoodi, Ph.D.

Academic Editor

PLOS ONE

Additional Editor Comments (optional):

Reviewers' comments:

Reviewer's Responses to Questions

**Comments to the Author**

1. If the authors have adequately addressed your comments raised in a previous round of review and you feel that this manuscript is now acceptable for publication, you may indicate that here to bypass the “Comments to the Author” section, enter your conflict of interest statement in the “Confidential to Editor” section, and submit your "Accept" recommendation.

Reviewer #2: All comments have been addressed

2. Is the manuscript technically sound, and do the data support the conclusions?

Reviewer #2: Yes

3. Has the statistical analysis been performed appropriately and rigorously? 

Reviewer #2: Yes

4. Have the authors made all data underlying the findings in their manuscript fully available?

Reviewer #2: Yes

5. Is the manuscript presented in an intelligible fashion and written in standard English?

Reviewer #2: Yes

6. Review Comments to the Author

Reviewer #2: (No Response)

7. PLOS authors have the option to publish the peer review history of their article (what does this mean?). If published, this will include your full peer review and any attached files.

Reviewer #2: No

---

## [Editor Report · Acceptance letter]

13 Jul 2023

PONE-D-22-31206R3 

Effects of turmeric (*Curcuma longa*) supplementation on glucose metabolism in diabetes mellitus and metabolic syndrome: an umbrella review and updated meta-analysis. 

Dear Dr. Jinatongthai:

I'm pleased to inform you that your manuscript has been deemed suitable for publication in PLOS ONE. Congratulations! Your manuscript is now with our production department. 

Kind regards, 

on behalf of

Dr. Mohammad Reza Mahmoodi 

Academic Editor

PLOS ONE